# Structural Significance of His73 in F-Actin Dynamics: Insights from Ab Initio Study

**DOI:** 10.3390/ijms231810447

**Published:** 2022-09-09

**Authors:** Tong Li, Juan Du, Mingfa Ren

**Affiliations:** 1Department of Engineering Mechanics, Dalian University of Technology, Dalian 116024, China; 2State Key Laboratory of Structural Analysis for Industrial Equipment, Dalian University of Technology, Dalian 116024, China

**Keywords:** ab initio, F-actin dynamics, His73, nucleotide exchange

## Abstract

F-actin dynamics (polymerization and depolymerization) are associated with nucleotide exchange, providing the driving forces for dynamic cellular activities. As an important residue in the nucleotide state-sensing region in actin, His73 is often found to be methylated in natural actin and directly participates in F-actin dynamics by regulating nucleotide exchange. The interaction between His73 and its neighboring residue, Gly158, has significance for F-actin dynamics. However, this weak chemical interaction is difficult to characterize using classic molecular modeling methods. In this study, ab initio modeling was employed to explore the binding energy between His73 and Gly158. The results confirm that the methyl group on the His73 side chain contributes to the structural stability of atomistic networks in the nucleotide state-sensing region of actin monomers and confines the material exchange (Pi release) pathway within F-actin dynamics. Further binding energy analyses of actin structures under different nucleotide states showed that the potential model of His73/Gly158 hydrogen bond breaking in the material exchange mechanism is not obligatory within F-actin dynamics.

## 1. Introduction

F-actin is polymerized from G-actin in living cells and acts as the structural basis for the mechanical performance and dynamic behaviors of microfilament networks [1]. F-actin polymerization and depolymerization are usually accompanied by nucleotide exchanges [2], providing free energy for various cellular activities. Due to the significance of the structure of actin on the spontaneous regulation of nucleotide exchanges in living cells, different actin nucleotide state-sensing models have been proposed based on in-vitro actin structural characterizations. However, determinant actin conformational changes during nucleotide exchange are still debatable [3,4]. R. Dominguez and co-workers reported that the ATP hydrolysis of F-actin would ultimately induce a loop-to-helix transition of a D-loop (residues 38–52, on subdomain 2) by a sequence of conformational changes, starting with the rotation of the Ser14 and His73 residues [5,6]. K. Trybus and co-workers later obtained ATP- and ADP-actin structures that have no significant conformational difference except for the bending of Glu72 and His73 on the sensor loop (H-loop, residues 70–78) [7]. Conformational changes to His73 between ATP- and ADP-actin structures were also reported by T. Wakabayashi recently [8]. These crystallographic pieces of evidence all indicate that the His73 conformational change plays a critical role in F-actin dynamics. Figure 1 provides the typical structural change of the H-loop in F-actin dynamics from the source of drosophila [7], where a significant bending deformation of Glu72 and His73 can be located.

In addition to conformational changes, the His73 residue on the H-loop also directly participates in nucleotide exchanges during F-actin polymerization. On natural actin monomers, His73 is the only residue that is often modified with an extra methyl, for which the biological function is still cryptic. According to in vivo experiments, mutations of His73 would significantly change the ATP hydrolysis rate [9]. The methylated His73 acts as a switch for Pi release in nucleotide exchange, which is an important material measurement for F-actin dynamics [9]. A cleft between subdomains 2 and 4 (refer to Figure 1) on actin was observed after ATP hydrolysis [10], creating the pathway for the material exchange. The molecular mechanism for the material-release pathway formation can be found in Figure 2. The chemical interactions between His73 and its neighboring residues, most importantly Gly158 [9,11], in the nucleotide state-sensing region are important for stabilizing the interaction between subdomains 2 and 4 and confining the Pi release pathway before F-actin depolymerization. It is, therefore, of fundamental significance to evaluate the binding energy between His73 and Gly158 for a better understanding of the role that His73 plays in regulating nucleotide exchanges in F-actin dynamics. Since this binding energy between His73 and Gly158 occurs due to weak chemical interactions that are difficult to characterize using classical molecular dynamics methodology, a higher-level physics calculation method is needed to accurately evaluate the binding energy, e.g., the ab initio method based on quantum mechanics theory.

In this study, we evaluate the stability of the nucleotide state-sensing region by analyzing the binding energy between His73 and Gly158 via ab initio modeling. Actin structures with different methylation and nucleotide states are studied to understand the structural significance of His73 conformational changes in nucleotide exchanges. The analysis provides insights into an improved understanding of the critical role of His73 in regulating nucleotide exchanges in F-actin dynamics.

## 2. Results and Discussion

### 2.1. The Structural Significance of His73 Methylation in F-Actin Dynamics

To theoretically validate the structural significance of His73 methylation in F-actin dynamics, six actin structures with different methylation states were considered in the present ab initio study, which were all in the ATP binding state. Using the method elaborated in Section 3, the binding energy between His73 and Gly158 was calculated for all cases and tabulated in Table 1. It is interesting to find that the binding energies in the three methylated actin structures are all lower than that in the corresponding unmethylated structures. During F-actin depolymerization, if the binding energy between His73 and Gly158 is relatively lower, actin monomers need to absorb more energy to separate subdomains 2 and 4 to create the pathway for material exchange (Pi release), which means a lower His73/Gly158 binding energy should result in the higher structural stability of the nucleotide state-sensing region. The positive binding energies between unmethylated His73 and Gly158 on the last three actin structures indicate that the unmethylated His73 results have relative structural instability. Figure 3 shows the His73/Gly158 configurations of these selected actin structures. It should be noted that the basis set superposition error (*E_bsse_*) is comparable to the binding energy (*E_b_*) characterization, confirming that the basis set superposition error is obligatory for the calculation of weak chemical interactions in biological structures.

The ab initio calculation results suggest that actin structures with side chain methyl groups all present higher structural stability. Nyman, et al. proposed that the side chain methylation of His73 would affect the interaction between His73 and its neighboring residues [11]. A typical mechanism for H-bond formation induced by His73 methylation is provided in Figure 4. When His73 is methylated, the side chain is fully substituted and carries one positive charge, which is shared between the two nitrogen atoms (δ1 and ε2, refer to Figure 3). This positive charge will contribute to enhancing the H-bond interaction between His73 and Gly158.

According to the ab initio characterization of the selected actin structures, the aforementioned H-bond (N-H···O) between His73 and Gly158 can be located on the actin structures 1ATN and 2A3Z (refer to Figure 3). However, as there is no H-bond between His73 and Gly158 on 1Y64 actin, the negative binding energy indicates that H-bond (N-H···O) is not the only source of the relatively lower binding energy between His73 and Gly158 on actin monomers with methylated His73. 

### 2.2. His73/Gly158 H-Bond Breaking Is Not Obligatory for Material Exchange in F-Actin Dynamics

The sequential conformational changes in actin monomers can trigger ATP hydrolysis and create a pathway for material exchanges in F-actin polymerization and depolymerization. In these conformational changes, the rearrangement of the H-loop after ATP hydrolysis putatively creates a pathway for Pi to release from the actin monomer after ATP hydrolysis [5,8]. To validate the molecular mechanism of the critical actin structure conformational changes during the nucleotide exchanges, we have designed ab initio modeling for another group of actin structures with different nucleotide-binding states, and the characterization results are provided in Table 2.

The binding energies between His73 and Gly158 on ATP-actin can be either higher or lower than ADP-actin based on the calculation of different actin structures. The conformations of His73/Gly158 are shown in Figure 5, with the molecular configurations for H-bond formation. According to the ab initio modeling of Oryctolagus actin structures, the His73/Gly158 binding energy is higher within the ATP state (1J6Z) than within the ADP state (1NWK). Instead of the relatively stronger nitrogen oxide H-bond hypothesis (N-H···O, refer to Figure 4), a weak carbon oxide H-bond (C-H···O) can be found on both ATP and ADP actin structures from Oryctolagus. Based on the ab initio modeling of Dictyostelium actin structures, the His73/Gly158 binding interaction is much higher in ADP actin (3A5L) than in ATP actin (3A5M). The His73/Gly158 binding interactions from within the ATP state to the ADP state for drosophila actin structures are close to each other, and both present a positive binding interaction (9.4~9.65 kJ/mol), which is due to the lack of side-chain methylation, as discussed in the last section.

It can be found that the ATP-actin structures do not necessarily experience an H-bond interaction between His73 and Gly158. ADP-actin can also present a highly stabilized His73/Gly158 interaction, which confines the material exchange pathway during F-actin polymerization and depolymerization. The molecular mechanisms for actin structure conformational changes in material exchange are still cryptic in biophysics analyses. More ab initio modeling of the interactions between other keynote residues concerning nucleotide-binding states should be developed to obtain more convincing evidence on the prominent conformation changing mechanisms which directly regulate nucleotide exchange processes and affect F-actin dynamics in cellular processes.

## 3. Materials and Methods

The interaction between the residues in the abovementioned actin structures is difficult to characterize using classic molecular dynamics simulation. Ab initio quantum chemistry methods can provide more information about the variation of weak interaction change in different states [12]. Freedman, et al., also used the ab initio modeling method to study the dynamics of ATP hydrolysis in Actin [13].

All the ab initio calculations were performed using the Gaussian09 package [14]. A Density Functional Theory (DFT)-level method B97D [15], with a basis set of 6-31G, was employed to accurately describe long-range dispersion during geometry optimization. Single point energy calculations are then performed at the Gaussian-2 (MP2/6-311++G**) level [16]. The basis set superposition error [17] was also calculated to guarantee the accuracy of the characterization of the weak chemical interactions. The binding energy (*E_b_*) between the His73 and Gly158 is defined as:(1)$$E_{b}=E_{comb}-E_{His73}-E_{Gly158}+E_{bsse}$$
where *E_His_*_73_, *E_Gly_*_158_, and *E_comb_* represent the energies of the isolated His73, Gly158, and the combined super-molecule, respectively, and *E_bsse_* denotes the basis set superposition error. The calculated binding energies between His73 and Gly158 were used to evaluate the stability of the atomistic networks of actin monomers with different nucleotide and His73 side chain methylation states. The molecular visualizations were carried out using VMD [18] and Avogadro [19] software.

To understand the role of His73 side chain methylation in nucleotide exchange in F-actin dynamics, six Oryctolagus skeletal muscle actin with different methylation states were first selected. Three of these crystal structures were of the methylated His73, including 1ATN [20], 1Y64 [21], and 2A3Z [22]. The other three actin structures were of unmethylated His73, including 2FF6 [23], 2V52 [24], and 3BUZ [25]. All of these six actin macromolecules were in the state of ATP binding. It should be noted that the six-protein structures are divided into two groups to reflect the influence of His 73 methylation in natural proteins. Therefore, no analysis homology structures were designed in this study.

An H-bond between His73 and Gly158 was reported to be important in stabilizing the molecular structures of actin monomers before ATP hydrolysis [8,11]. To validate the source of the critical interaction between His73 and Gly158 in F-actin dynamics, another three groups of representative actin structures with different nucleotide states were also selected for ab initio modeling. The molecular structures in each actin group can be independently determined from both ATP- and ADP-actin monomers under similar laboratory conditions and are provided in Table 3. It should be noted that the His73 side chains on 2HF3 and 2HF4 are unmethylated. 

## 4. Conclusions

In the present study, we employed the ab initio method to investigate the structural significance of His73 in F-actin dynamics. The ab initio characterization of the binding energy between His73 and Gly158 in actin monomers suggests that the methyl group on the His73 side chain contributes to the structural stability of atomistic networks in the nucleotide state-sensing region of actin monomers and confines the pathway of material exchange in F-actin dynamics. However, the potential His73/Gly158 H-bond breaking mechanism for the formation of the Pi release pathway is not obligatory during F-actin depolymerization. As a quantum mechanics-level method, ab initio modeling is the first method ever employed to understand the structural significance of His73 in F-actin dynamics and shows great potential to understand various structural basis findings associated with dynamic cellular processes.

## Figures and Tables

**Figure 1 ijms-23-10447-f001:**
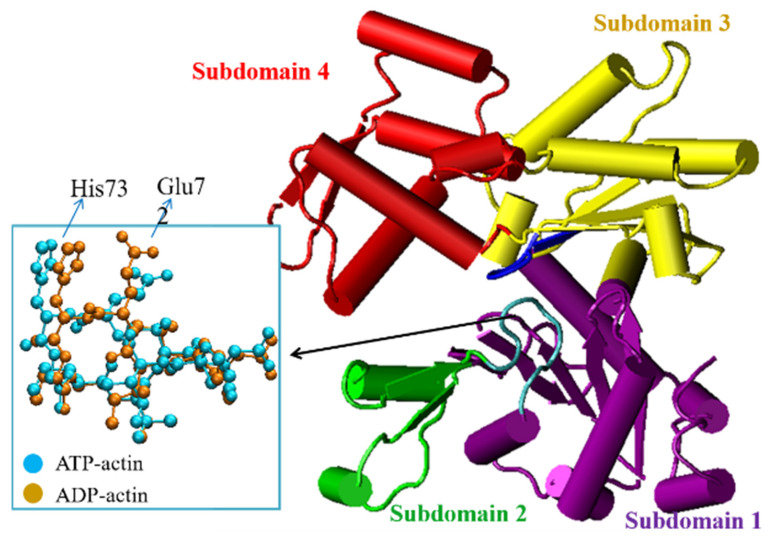
The atomistic structures of a H-loop on actin in both ATP and ADP states. The corresponding IDs of the PDB sources are 2HF4(ATP) and 2HF3(ADP) [7].

**Figure 2 ijms-23-10447-f002:**
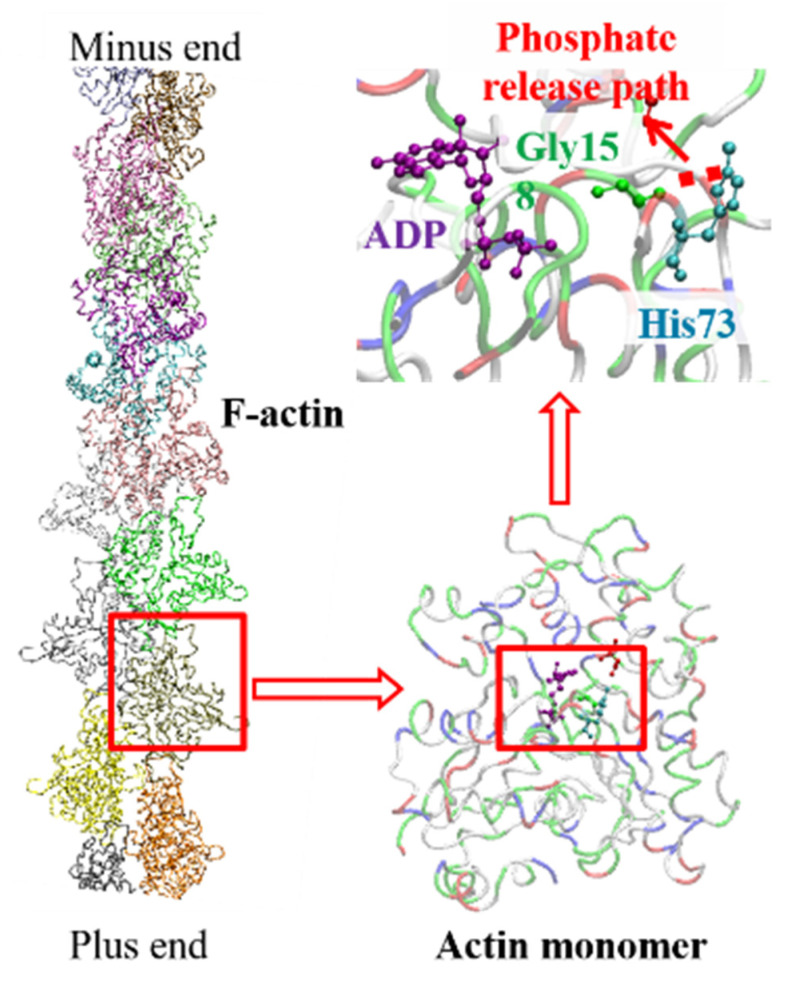
The structural basis for material exchange in F-actin dynamics. The interactions between His73 and Gly158 [8,11] were reported to be important in stabilizing the atomistic networks on actin. The connections between these residues are disrupted during the nucleotide exchange, which helps to create the pathway for material exchange. Once the Pi releases, its mother globular actin would gradually reach the minus end of the actin filament and depolymerize from F-actin.

**Figure 3 ijms-23-10447-f003:**
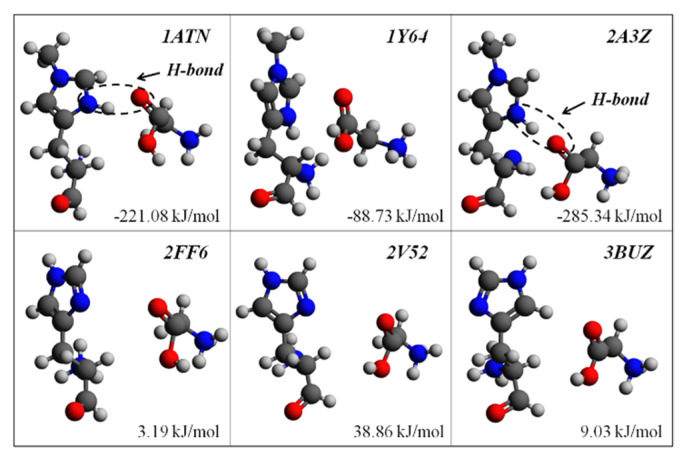
Conformations of the six His73/Gly158 interactions. The structures of the first three (upper row) are of methyl groups on the side chain. H-bonds (N-H···O) can be found on 1ATN and 2A3Z but not on 1Y64. The binding energies on the unmethylated structures (lower row) are all above zero, which would result in the structural instability of the nucleotide sites on actin. Color key: O, red; C, gray; N, blue; H, silver.

**Figure 4 ijms-23-10447-f004:**
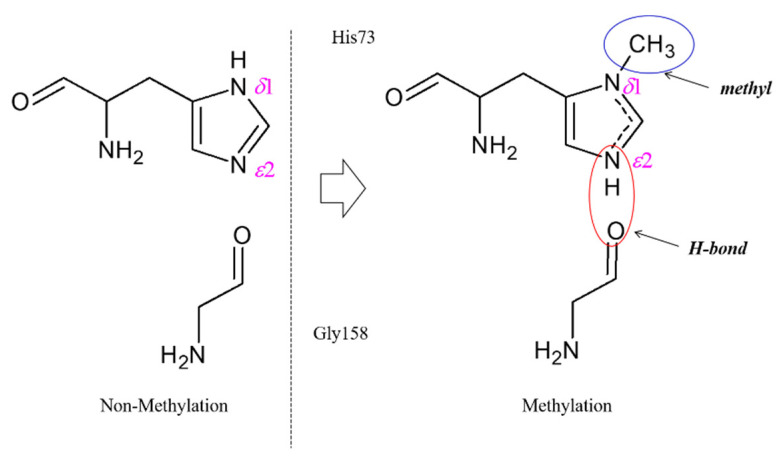
The interaction between His73 and Gly158 could be enhanced by the formation of an H-bond when His73 experiences side chain methylation.

**Figure 5 ijms-23-10447-f005:**
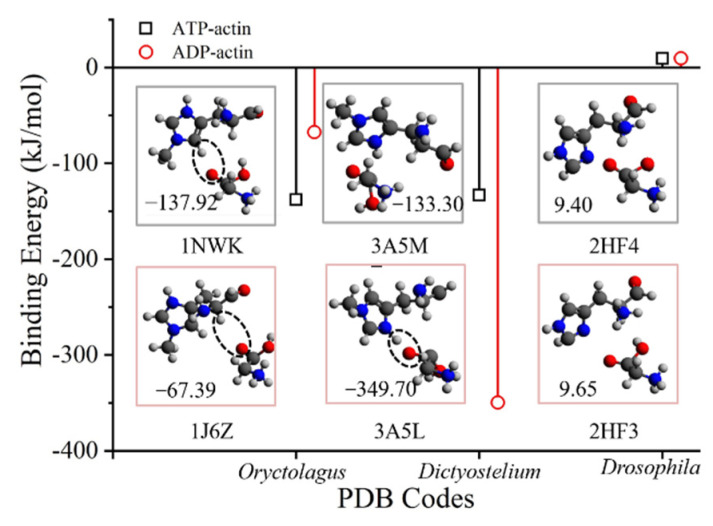
The weak chemical interaction between His73 and Gly158 for both ATP and ADP-actin. The binding energies for ATP-actin are not obligatorily lower than those for ADP-actin. H-bonds (dash circle) could not be found on all ATP-actin structures (PDB codes: 3A5M and 2Hf4), yet it appears in some ADP-actin structures (PDB codes: 1J6Z and 3A5L). The values in the conformation pictures denote the binding energy of the corresponding actin structure. Color key: O, red; C, gray; N, blue; H, silver.

**Table 1 ijms-23-10447-t001:** His73/Gly158 binding energy calculation results in different methylation states.

PDB ID	States ^1^	Ecomb ^2^	E_His73_	E_Gly158_	E_bsse_	E_b_
1ATN	ATP/Y	−2,111,763.69	−1,357,798.71	−753,724.23	17.43	−221.08
1Y64	ATP/Y	−2,111,713.21	−1,357,771.33	−753,839.34	12.91	−88.73
2A3Z	ATP/Y	−2,111,797.72	−1,357,715.98	−753,774.68	18.84	−285.34
2FF6	ATP/N	−2,007,737.54	−1,253,918.49	−753,814.92	7.35	3.19
2V52	ATP/N	−2,007,744.46	−1,253,947.10	−753,829.20	7.41	38.86
3BUZ	ATP/N	−2,007,754.82	−1,253,882.65	−753,872.34	8.95	9.03

^1^ The states of actin include the nucleotide states (ADP or ATP) and side-chain methylation (Yes or No). ^2^ All the energy values are in the unit of kJ/mol.

**Table 2 ijms-23-10447-t002:** His73/Gly158 binding energy calculation results in different nucleotide-binding states.

PDB ID	States ^1^	Ecomb ^2^	E_His73_	E_Gly158_	E_bsse_	E_b_
1NWK	ATP/Y	−2,111,727.89	−1,357,765.61	−753,810.78	12.18	−137.92
1J6Z	ADP/Y	−2,111,661.74	−1,357,758.88	−753,828.18	6.61	−67.39
3A5M	ATP/Y	−2,111,746.51	−1,357,783.93	−753,814.66	13.26	−133.30
3A5L	ADP/Y	−2,111,751.50	−1,357,775.95	−753,597.98	24.34	−349.71
2HF4	ATP/N	−2,007,709.88	−1,253,871.19	−753,841.34	6.85	9.40
2HF3	ADP/N	−2007699.82	−1253870.48	−753830.77	8.32	9.65

^1^ The states of actin include the nucleotide states (ADP or ATP) and side-chain methylation state (Yes or No). ^2^ All the energy values are in the unit of kJ/mol.

**Table 3 ijms-23-10447-t003:** The structural information of the selected actin structures in Pi release mechanisms analysis.

PDB ID	Nucleotide State	Methylation	Sources	Reference
1NWK	ATP	Yes	*Oryctolagus*	Dominguez, et al. 2001, 2003 [5,6]
1J6Z	ADP	Yes		
3A5M	ATP	Yes	*Dictyostelium*	Murakami, et al. 2010 [8]
3A5L	ADP	Yes		
2HF4	ATP	No	*Drosophila*	Rould, et al. 2006 [7]
2HF3	ADP	No		

## Data Availability

The data presented in this study are available in this paper.

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
