# Peer review of "Structural Significance of His73 in F-Actin Dynamics: Insights from Ab Initio Study"

_ijms, 2022, doi:10.3390/ijms231810447_

Round 1

Reviewer 1 Report

In this work by Tong Li, Juan Du and Mingfa Ren entitled “Structural Significance of His73 in F-actin Dynamics: Insights from Ab Initio Study”, the authors evaluate the nucleotide state-sensing region stability between His73 and Gly158 via ab initio modeling.  Actin structures with different methylation and nucleotide states were studied, using modelling calculations to appreciate whether the His73 conformational changes in nucleotide exchanges has an impact on overall structure. The work is of interest and offers some data from a model based analysis. There is however no direct evaluation of whether the modelling used is appropriate and whether the data presented would actually be correct. Some validations of the data would certainly strengthen and support their analysis

In some of the analysis and the modelling, measurements are made using crystallographic structures obtained from different species. Given the numerous differences seen between Dyctyostelium and Drosophila actin, is it even possible and appropriate to analyse these structures via modelling, as numerous different parameters could explain the differences seen?

Other points to consider:

The authors refers to His73 methylation numerous times in the work but never explain the biological implication of this posttranslational modifications.

The authors do not keep an accurate number on the figures presented. Figures 1 and 2 are duplicated and there are no figures 3-4 but one figure 5. However figures 3 and 4 are referred to in the text but it is not clear what these are?The authors should be urged to pay more attention to their work prior to make it available for others to read.

Author Response

Dear Reviewer,

Thank you for your valuable comments. We have responded to your questions, please see the attachment.

Reviewer 2 Report

Actin is a family of multifunctional proteins playing a pivotal role in maintaining the cytoskeleton. The free monomeric actin is called G-actin, and the linear polymeric filamentous actin is called F-actin. G-actin also has one ATP binding site per monomer. The polymerization and depolymerization of F-actin from G-actin are driven by nucleotide exchange. The energy released during this step is required for cellular dynamic activities. The residue His73 on the monomeric actin is methylated and interacts with neighboring Gly158. This region is termed as nucleotide state sensor region on actin. In the current manuscript, the authors have generated ab-initio models to characterize the interaction mechanism between methylated His73 and Gly128. The study revealed that the methyl group on His73 contributes to the structural stability of atomistic networks in the nucleotide state-sensing region of actin monomers and confines the pathway for material exchange in F-actin dynamics.

The draft is well written and provides considerable graphics details. The present work matches the journal's scope. I do not have significant experimental comments to address, but minor comments must be addressed before an editorial decision.

Comments:

1.     The structures of the actin are available in the database, RCSB; then why did the authors generate ab-initio models, and why were homology models not used in the study to address the mechanism of interaction? This need to be discussed with an explanation in the text.

2.     In the introduction paragraph, the authors did not provide details about the source of actin protein?

Minor comments-

The font in Figures: It appears that the label font is not consistent in the figures. Also, the figure size can be Improved.

Page 1, lines 43-44: "Figure 1 provides a typical……" This sentence is not clear to me. The authors can rephrase it.

Page2, line 49: "directly participates nucleotides exchanges during." The sentence can be written here "directly participates in nucleotides exchanges during."

Page 3, line 93: Correct the formatting of "conFigurations." Similarly, correct the same in line 138 on page 4.

Page 5, line 152: The PDB codes of the structures are mentioned here; however, the term "PDB code" is missing. It is better to mention the PDBs with the term "PDB codes or PDB accession etc." Also, correct the formatting of the two codes in line 152, page 5.

Page 6, line 198, and line 204: In some places, the term "ab initio" is written in italics and as regular. Be consistent.

Page 8, lines 260 and 267: The journal names are inconsistent in the reference list.

Author Response

Dear Reviewer,

Thank you for your valuable comments. We have replied to your questions. Please refer to the attachment.
